# Takotsubo Syndrome during COVID-19 Pandemic in the Veneto Region, Italy

**DOI:** 10.3390/v14091971

**Published:** 2022-09-06

**Authors:** Marco Zuin, Giacomo Mugnai, Maurizio Anselmi, Stefano Bonapace, Paolo Bozzini, Fabio Chirillo, Ada Cutolo, Giuseppe Grassi, Daniela Mancuso, Samuele Meneghin, Giulio Molon, Antonio Mugnolo, Ivan Pantano, Angela Polo, Paola Purita, Loris Roncon, Salvatore Saccà, Daniele Scarpa, Domenico Tavella, Sakis Themistoclakis, Giovanni Turiano, Roberto Valle, Maddalena Widmann, Edlira Zakja, Alberto Zamboni, Gianluca Rigatelli, Claudio Bilato

**Affiliations:** 1Division of Cardiology, West Vicenza General Hospitals, Via del Parco 1, 36071 Arzignano-Vicenza, Italy; 2Department of Cardiology, Fracastoro Hospital, 37020 San Bonifacio-Verona, Italy; 3Department of Cardiology, IRCCS Sacro Cuore Don Calabria Hospital, 37024 Negrar-Verona, Italy; 4Department of Cardiology, San Bassiano Hospital, 36061 Bassano-Vicenza, Italy; 5Department of Cardiology, All’Angelo Hospital, 30174 Mestre-Venezia, Italy; 6Department of Cardiology, Santi Giovanni & Paolo Hospital, 30122 Venezia, Italy; 7Department of Cardiology, Padua University Hospital, 35100 Padova, Italy; 8Department of Cardiology, Legnago General Hospital, 37045 Legnago-Verona, Italy; 9Department of Cardiology, Chioggia General Hospital, 30015 Chioggia-Venezia, Italy; 10Department of Cardiology, Mirano General Hospital, 30035 Mirano-Venezia, Italy; 11Department of Cardiology, Rovigo General Hospital, 45100 Rovigo, Italy; 12Department of Cardiology, Verona University Hospital, 37100 Verona, Italy; 13Department of Cardiology, San Donà General Hospital, 30027 San Donà di Piave-Venezia, Italy; 14Department of Cardiology, Madre Teresa di Calcutta Hospital, 35043 Schiavonia-Padova, Italy

**Keywords:** Takotsubo, COVID-19, epidemiology

## Abstract

**Background:** During the COVID-19 pandemic, the risk of SARS-CoV-2 infection, the public health measures of social distancing, the freedom limitations, quarantine, and the enforced homeworking under the lockdown period, as well as medical causes including COVID-19 infection per se, may have caused major emotional distress, especially in the most vulnerable patients. We aimed to evaluate the variations in the number of admissions due to Takotsubo syndrome (TTS) during the COVID-19 pandemic in the Veneto region. **Methods:** We retrospectively reviewed and analyzed the number of admissions because of TTS in 13 Divisions of Cardiology located in the Veneto region, the northeastern area of Italy, covering a population of more than 2.5 million inhabitants, during the two major pandemic waves of COVID-19 (the first between 15 March and 30 April 2020 and the second between 15 November and 30 December 2020) that occurred in 2020. **Results:** In total, 807 acute coronary syndromes were admitted in the 13 enrolling hospitals. Among these, 3.9% had TTS. Compared to the corresponding 2018 and 2019 time periods, we observed a significant increase in the number of TTS cases (+15.6%, *p* = 0.03 and +12.5%, *p* = 0.04, comparing 2018 to 2020 and 2019 to 2020, respectively). Geographical distribution of the TTS cases reflected the broad spread of the SARS-CoV-2 infection with a significant direct relationship between TTS incidence and the number of COVID-19 infections according to Pearson’s correlation (r = 0.798, *p* < 0.001). **Conclusions:** The higher incidence of TTS during the 2020 COVID-19 pandemic waves, especially in the areas that were hit hardest in terms of morbidity and mortality by the SARS-CoV-2 infection, suggest a strong direct and/or indirect role of COVID-19 in the pathogenesis of TTS.

## 1. Introduction

The Veneto region, which is in the northeast of Italy, was hit hard by the COVID-19 infection during the two major 2020 COVID-19 pandemic waves. Indeed, the first death due to SARS-CoV-2 infection in western countries occurred in the municipality of Vò Euganeo, a small town near Padua [1]. According to recent analyses [2], a rise in the incidence of Takotsubo syndrome (TTS) has been observed worldwide during the COVID-19 pandemic compared to pre-pandemic periods [2]. TTS is defined as a reversible acute heart dysfunction, generally triggered by emotional or physical stress, having a clinical presentation that resembles acute coronary syndrome. Whether this increase in TTS occurrence might be due to an increased psychological and emotional burden because of quarantine, to the lack of social interactions and community relationships, to economic consequences produced by the prolonged lockdown periods or because of medical causes, including COVID-19 infection per se, remains unknown [2,3,4]. The aim of the present study is to analyze the clinical characteristics and trends of patients with TTS admitted at the Divisions of Cardiology of 13 hospitals of the Veneto region during the first and second pandemic waves in 2020 compared to the corresponding 2019 and 2018 time intervals.

## 2. Materials and Methods

### 2.1. Study Design and Patients

This is a multicenter, retrospective study including all the consecutive patients presenting with TTS, hospitalized in 13 different hospitals of the Veneto region covering a population of 2,554,818 inhabitants (Appendix A) during the first (between 15 March 2020 and 30 April 2020) and second (between 15 November 2020 and 30 December 2020) COVID-19 pandemic waves (the 2020 cohort). Conversely, TTS patients admitted over the same time windows in 2019 and 2018 were considered as the historical cohorts. Reverse transcriptase-polymerase chain reaction (RT-PCR) for SARS-CoV-2 on nasopharyngeal swabs was performed on admission in all the patients. Inclusion criteria were: all the consecutive patients, aged ≥18 years old, referred and hospitalized for TTS in one of the 13 enrolling hospitals in the Veneto region irrespective of their hemodynamic conditions. All the patients aged <18 years, or without a confirmed diagnosis of TTS, or admitted because of other causes of cardiac chest pain were excluded. Per study protocol, TTS was diagnosed according to the InterTak criteria [5]. For all patients, medical history, clinical examination, 12-lead ECG, left ventricular ejection fraction (LVEF), arrhythmic events, and cardiac magnetic resonance (CMR) data (if performed) were collected and compared to the historical cohort. Two trained cardiologists, who were unaware of the design of the study, independently reviewed and extracted the local (at each recruiting center) clinical and instrumental records. Data were then managed and analyzed at the coordinating center by two trained physicians. Ethical approval was obtained from the Ethical Committee of the coordination centre (Division of Cardiology, West Vicenza General Hospitals).

### 2.2. Aims of the Study

The primary aim of the study was to analyse the clinical characteristics and trends of patients with TTS admitted during the first and second pandemic waves compared to the corresponding 2019 and 2018 periods. Secondary goals included the comparison of the LVEF and the mortality rate between the two groups.

### 2.3. Statistical Analysis

Continuous variables were expressed as mean ± standard deviation (SD) or interquartile range (IQR) and were compared by Student’s *t*-test if the data had normal distribution, otherwise by Wilcoxon–Mann–Whitney U test. Categorical variables were presented as proportions and compared by Pearson’s χ2 test. Pearson’s correlation was carried out to evaluate the presence of a potential association between the new TTS cases and the number of COVID-19 infections. The crude incidence (×100.000 inhabitants) of TTS was also calculated as the ratio of number events and mid-year population × 100.000.

Poisson regression was used to estimate the annual percentage change (AAPC) in TTS events with 95% confidence interval (CI). Specifically, the dependent variable was the number of events, while the independent variables were weekly and annual seasonality of acute coronary syndromes (ACS) hospitalizations. The AAPC was defined as −100 × [1 − exp(β)] where β represents the coefficient for calendar year. Furthermore, the model was checked for overdispersion and, if any, a dispersion parameter was included in the model.

Statistical significance was defined as *p* < 0.05. Statistical analyses were performed using SPSS package version 20.0 (SPSS, Chicago, IL, USA).

## 3. Results

### 3.1. Incidence of TTS

Over the study period, during the first and the second pandemic waves, 807 acute coronary syndromes were admitted in the 13 recruiting hospitals. Among these, 32 patients (3.9%) had a TTS. The cumulative number of TTS cases observed in the first and second COVID-19 pandemic waves in 2020 and in the historical cohorts, are shown in Figure 1. Compared to the historical cohorts, a significant increase in the number of TTS cases occurred in 2020 (from to 27 to 32, +18.5%, *p* = 0.029 and from 28 to 32, +14.3%, *p* = 0.039, comparing 2018 to 2020 and 2019 to 2020, respectively) (Figure 1A). These findings were confirmed by the increased crude incident rate observed in 2020 (first plus second pandemic waves periods) (Figure 1B). Intriguingly, the areas with higher TTS incidence (Provinces of Venice, Verona, and Padua) were those with the larger spread of the SARS-CoV-2 infection and the higher number of COVID-19 patients (n = 134,914 patients at the end of the study period) (Figure 2). In this regard, Pearson’s correlation revealed a significant direct relationship between the TTS cases and the number of COVID-19 infections (r = 0.798, *p* < 0.001) (r = 0.71, *p* = 0.001, r = 0.821, *p* < 0.0001, and r = 0.792, *p* < 0.0001 for the provinces of Padua, Verona, and Venice, respectively).

A significant age-adjusted increase in TTS event rates (+1.75%/year, 95% CI: 1.72–1.78, *p* = 0.006), especially in women (+2.2%/year, 95% CI: 2.0–2.4, *p* = 0.001), was observed during the study period, as shown in Table 1.

### 3.2. Clinical Characteristics

The general characteristics of the population analyzed are presented in Table 2. During the first (n = 10) and second (n = 22) COVID-19 pandemic waves, 32 patients (mean age 73.5 ± 11.7 years, 30 females) were admitted with a diagnosis of TTS. Of these, three subjects (9.4%) were positive for SARS-CoV-2. The 70–79 years age group was the most affected by TTS both in the 2020 and in the historical cohorts. Age, gender, prevalence of dyslipidemia, TTS triggers, LVEF, symptoms at admission, electrocardiographic features, presence of coronary artery disease, and onset modalities were similar between the two groups. Arterial hypertension was more frequent in the 2020 cohort (81.2% vs 49.1%, *p* = 0.04), while the prevalence of diabetes mellitus was higher in the historical cohorts. Interestingly, the 2020 cohort showed normal CRM features less frequently (76.1% vs 93.4%, *p* = 0.041) and, notably, all the three COVID-19 patients with TTS had a pathological CMR. Cardiac troponins (T or I) were higher in the 2020 cohort (*p* = 0.003 and 0.002, respectively), including the three patients with concomitant COVID-19.

### 3.3. Outcomes

The global rate of hospital readmission observed in 2020, defined as the sum of the cases observed during the two pandemic waves in that year, as well as the overall mortality rate in the same period were similar when compared to the historical cohorts (3.1% vs. 9.1%, *p* = 0.282 and 0% vs. 3.6%, *p* = 0.296, respectively). The total in-hospital stay was shorter in 2020 compared to the historical cohorts (*p* = 0.021).

## 4. Discussion

To our knowledge, this is the first study to systematically investigate the incidence and the clinical characteristics of TTS patients during the first and second pandemic waves among a population of more than 2 million inhabitants [6]. The principal finding of our analysis is the increasing incidence of TTS during the pandemic compared to control groups. Our findings are in line with those reported by Shah et al. [7], demonstrating an increased incidence of TTS in both the general population and COVID-19 patients probably related to the generalized increases in psychological distress, the cytokine storm, increased sympathetic responses in COVID-19 patients, and microvascular dysfunction. Moreover, our observed TTS pre-pandemic incidence is comparable to that presented by Jabri et al. [2].

Notably, the number of TTS cases were higher during the second pandemic wave, probably reflecting the effects of the previous lockdown, the prolonged restrictive measures, and the daily burden of the COVID 19-related economic consequences. In this regard, it is noteworthy that the higher incidence was observed in the areas where the infection hit harder during the first wave of the pandemic. Moreover, as pointed out elsewhere [6], by the end of 2020 (second wave) the Veneto region showed a drastic increase in SARS-CoV-2 infections compared to the first wave with a significantly higher COVID-19 spread and severity (CoSS) index compared to other Italian regions (23.1 vs. 6.3. of Lombardy, 4.1 of Piedmont, and 4.6 of Valle d’Aosta) due to the diverse regimen of the applied restrictions. Conversely, the variation in TTS observed compared with 2018 and 2019 must be interpreted as a natural variation in the incidence, which is commonly observed in many diseases, as also demonstrated by the lack of a statistically significant difference between the two years.

Laboratory findings revealed higher serum troponin levels in patients admitted during the COVID-19 pandemic compared to historical cohorts, probably because of a delayed presentation of the TTS patients due to the fear of infection, when troponin had already peaked. However, despite the deferred admission, no fatal events during the hospitalization were observed in the 2020 cohort. Our results agree with the findings provided by Jabri et al. [2] who also described no differences in the readmission rate of TTS patients hospitalized during the pandemic compared to the pre-pandemic period.

Compared to historical cohorts, 2020 TTS patients showed a significantly shorter in-hospital stay, probably reflecting an indirect effect of the COVID-19 pandemic [2]. During the pandemic waves, national and regional public health systems struggled with the possible overload of the hospital’s capacity and the risk of the infection spreading into “clean” areas of the hospitals [8]. As consequence, patients were probably discharged as soon as possible. Moreover, we cannot exclude that the observed TTS incidence may be underestimated considering the overall decline in admissions for acute coronary syndromes and, in general, for chest pain during the pandemic.

Among the patients affected by TTS during the COVID-19 pandemic waves, those aged 70–79 years were the most represented. Interestingly, individuals of the same age were at the highest risk of COVID-19 mortality, with this group having the highest number of cumulative deaths in Northern Italy due to COVID-19 [9]. Whether this result, by direct or undirect mechanisms, was related to the highest incidence of TTS in 70–79-year-old subjects remains to be clarified, but it is interesting to note that we identified in our retrospective analysis three patients with TTS and concomitant COVID-19. All these patients showed pathological findings of TTS at CMR. This aspect might be relevant because, to date, only a few cases have been reported on the occurrence of TTS in SARS-CoV-2-infected patients [10,11] and the mechanisms involved in the pathophysiology of TTS in COVID-19 patients have not yet been identified.

Nationwide, a significant reduction in hospitalizations for myocardial infarction in Italy in the COVID-19 era was reported [12]. Accordingly, a similar trend was observed in the Veneto region, especially during the first pandemic wave [12]. In particular, the latest epidemiological data have reported 120 ACSs/100,000 residents per year in our region; in particular, 6,648 and 6,848 patients were hospitalized for ACS in 2018 and 2019, respectively [13]. Moreover, among our study population of more than 2.5 million inhabitants, the hospital admissions for acute coronary syndrome (ACS) decreased by 37% from the year 2018 to 2020 and by 22% from 2019 to 2020, with a comparable trend when hospitalizations for all acute cardiovascular conditions were considered. Therefore, considering the general decline in hospital admissions during the pandemic waves, the percentage increase in TTS observed in our 2020 population compared to the historical cohorts is probably underestimated. Indeed, the current literature reports an incidence of TTS ranging from 1.0% to 2.0% in patients presenting with ACS [14], although it remains difficult to assess the incidence of this syndrome in the general population. However, we cannot exclude that the real incidence of ACS as well as TTS may have been higher considering that some patients could have died for these acute cardiovascular diseases without seeking medical attention during the pandemic [15].

Furthermore, it must be considered that the manifestation of TTS may also be different among gender. In fact, this syndrome is generally more common in females but about one male out of four presents with in-hospital complications, which may affect the outcome [16].

A certain fluctuation in the incidence of a disease is reasonable when comparing consecutive years, especially if some affecting factors, such as seasonality and climatic factors are implicated in its epidemiology [17,18]. Moreover, regarding the increased number of TTS cases and relative rates during the pandemic, it should be noted that our data are certainly underestimated. Indeed, during the outbreak, patients with ACS or chest pain often avoided seeking medical attention due to the fear of becoming infected. Unfortunately, this behavior determined an incorrigible sampling bias.

### Limitations

The major limitations of our hypothesis-generating study are its retrospective nature and the relatively few numbers of patients enrolled in the analysis. However, it should be noted that our observations were obtained from 13 Divisions of Cardiology of 13 different hospitals serving more than 2.5 million inhabitants and covering at least half of the total population of the Veneto region. Secondly, because of the major reorganizations of the public health system with the conversion of several general hospitals to COVID facilities following the pandemic [19], we cannot exclude that some patients, especially those living at the borders of our area of observation, were referred to hospitals outside the region, with the potential to partially alter our results. We cannot evaluate the incidence of TTS as a continuous timeline period since the pandemic waves were limited over time. Therefore, we considered two separate temporal windows of 45 days; this obligatory choice may have distorted our results, which may have been conditioned by randomness typical of the trend and seasonality of historical data. Moreover, the observed trend in 2020 compared to the previous years might be suggestive of an increase in case detection. However, to our knowledge, TTS incidence across the years has been unknown in Italy, and, in addition, all European countries (and Italy at the most) were hit by the pandemic. Therefore, no “true” reference is available. Finally, we cannot exclude that the observed small difference could be considered as a random difference around the true mean of TTS hospitalization cases.

## 5. Conclusions

The present multicentric study, based on a population of more than 2.5 million inhabitants, demonstrates a higher incidence of TTS during the 2020 COVID-19 pandemic waves compared to the same periods of the previous 2 years, especially in the areas that were hit hardest, in terms of morbidity and mortality, by the SARS-CoV-2 infection during the first wave of the pandemic. These observations suggest a strong direct and/or indirect role of COVID-19 in the pathogenesis of TTS. However, our results must be confirmed by dedicated larger and multicentric studies.

## Figures and Tables

**Figure 1 viruses-14-01971-f001:**
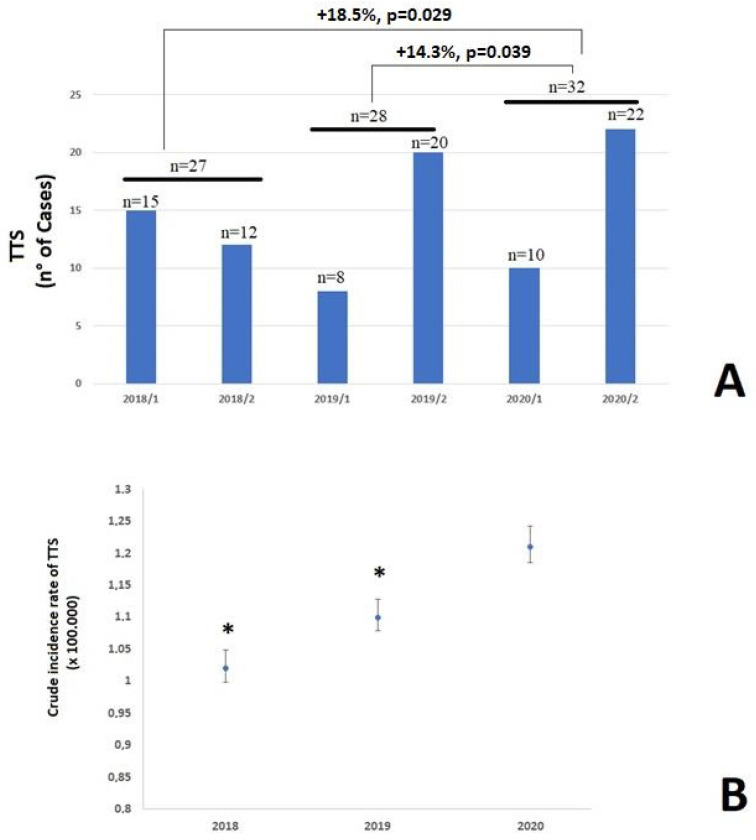
(**A**) Trends in Takotsubo syndrome hospital admission in Veneto region. Periods: 2018/1: between 15 March and 30 April 2018; 2018/2: between 15 November and 30 December 2018; 2019/1: between 15 March and 30 April 2019; 2019/2: between 15 November and 30 December 2019; 2020/1: between 15 March and 30 April 2020; 2020/2: between 15 November and 30 December 2020. (**B**) Crude incidence of Takotsubo syndrome over the years in Veneto region (per 100,000 inhabitants). * *p* < 0.05 compared to 2020; p for difference between 2018 vs 2019 = 0.621.

**Figure 2 viruses-14-01971-f002:**
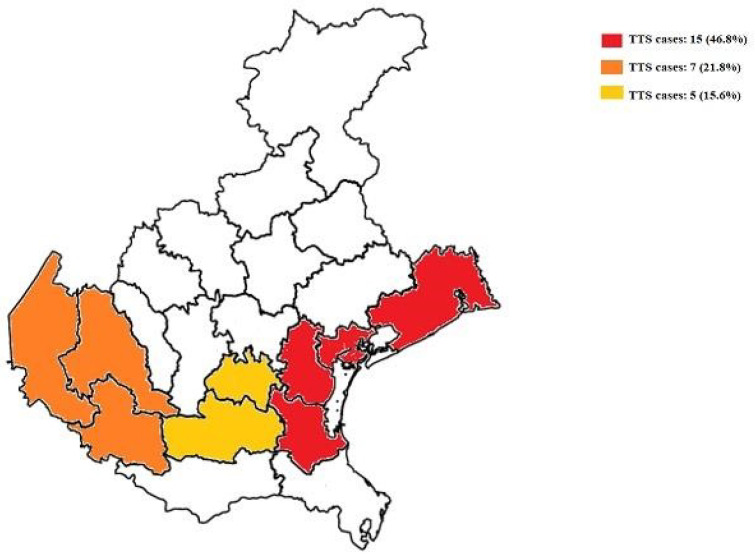
Map of the Veneto region showing the areas with the highest incidence of Takotsubo syndrome during the first and second pandemic waves.

**Table 1 viruses-14-01971-t001:** Trends in Takotsubo (TTS) event rates also stratified by gender between 2018 and 2020. CI: confidence interval.

	Annual Percentage Change (95% CI)	*p*
Overall	+1.75 (1.72–1.78)	0.006
Men	+1.3 (1.1–1.4)	0.02
Women	+2.2% (2.0–2.4)	0.001

**Table 2 viruses-14-01971-t002:** Baseline characteristics, clinical presentation, and outcomes.

Variables	2020 Cohort N = 32	Historical Cohort (2018–2019) N = 55	*p*
Age (years)	73.5 ± 11.7	69.2 ± 11.1	0.941
Age ≥65 years (%)	27 (84.4)	39 (70.9)	0.163
Female, n (%)	30 (93.8)	48 (87.3)	0.332
Arterial hypertension, n (%)	26 (81.2)	27 (49.1)	0.042
Diabetes mellitus, n (%)	1 (3.1)	8 (14.5)	0.032
Dyslipidemia, n (%)	14 (43.8)	28 (50.9)	0.527
**TTS triggers**			
None	11 (34.4)	15 (27.3)	0.488
Emotional	14 (43.8)	19 (34.5)	0.396
Physical	6 (18.6)	17 (30.9)	0.217
Emotional and physical	1 (3.1)	4 (7.3)	0.422
Positive COVID-19 (at the admission for TTS), n (%)	3 (9.4)	-	-
**Presenting symptoms**			
None, n (%)	1 (3.1)	3 (5.5)	0.606
Chest pain, n (%)	24 (75.0)	38 (69.1)	0.553
Dyspnea, n (%)	5 (15.6)	9 (16.4)	0.925
Syncope, n (%)	3 (9.4)	5 (9.1)	0.967
**ECG Features**			
Normal, n (%)	9 (29.0)	10 (18.2)	0.247
ST-segment elevation, n (%)	10 (32.3)	20 (36.4)	0.705
ST-segment depression, n (%)	2 (6.5)	4 (7.3)	0.882
Negative T waves, n (%)	16 (32.3)	21 (38.2)	0.586
Mean length of stay (days)	5.4 ± 2.3	8.5 ± 2.8	0.021
**Coronary Angiography**			
Significant coronary artery disease	2 (6.2)	6 (10.9)	0.464
Troponin T, ng/L	325 (51–1278) *	76 (29–586) **	0.003
Troponin I, ng/L	675 (126–3675) ^§^	345 (38–3276) ^§§^	0.002
LVEF (%)	49.6 ± 10.3	49.4 ± 8.7	0.918
**Arrhythmic events, n (%)**			
None, n (%)	29 (90.3)	47 (92.2)	0.766
AF, n (%)	2 (6.5)	2 (3.9)	0.586
NSVT, n (%)	1 (3.2)	0	0.216
SVT, n (%)	2 (3.9)	0	0.166
VF, n (%)	0	0	-
2nd or 3rd AV blocks, n (%)	0	0	-
**Cardiac Magnetic Resonance**			
Normal, n (%)	16/21 (76.1)	43/46 (93.4)	0.041
Edema, n (%)	3 (14.2)	2 (4.3)	0.152
LGE, n (%)	2 (9.5)	1 (2.1)	0.172
**Outcomes**			
Re-hospitalization, n (%)	1 (3.1)	5 (9.1)	0.282
In-hospital mortality, n (%)	0	2 (3.6)	0.296

TTS: Takotsubo syndrome; LVEF: left ventricular ejection fraction: AF: atrial fibrillation; NSVT: non-sustained ventricular tachycardia; SVT: sustained ventricular tachycardia; VF: ventricular fibrillation; LGE: late gadolinium enhancement. * Data available for 12 patients; ** data available for 24 patients. ^§^ data available for 10 patients; ^§§^ data available for 26 patients.

## Data Availability

Data are available, upon reasonable request, by contacting the corresponding author.

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
