# Peer review of "Takotsubo Syndrome during COVID-19 Pandemic in the Veneto Region, Italy"

_viruses, 2022, doi:10.3390/v14091971_

Round 1

Reviewer 1 Report

In this article authors have described the increased in  TTS during the COVID 19 outbreak. I have the following questions.

- Describe what is the baseline ACS admission rates in northen Italy. Consider referring to this article that discussed about reduction in ACS admissions during COVID 19 from Northen Italy, implying that the real numbers could be higher "N Engl J Med 2020; 383:88-89  DOI: 10.1056/NEJMc2009166"

- In the introduction mention the hypothesis, did the authors want to study change in the pattern of TTS during COVID 19. 

- Why did the authors compare 2020 to 2019+2018 ? If not so clarify

- Include inclusion and exclusion criteria for the study participants

- How was data extracted? Each hospital vs Private data base, Who extracted the data? Trained health care workers vs insurance/billing companies. Who analyzed the data (Statistician vs trained physician)

- 32 is 3.9 % for total ACS of 807 not 4.4 %

- How can the authors say the numbers were significantly high without establishing prior sample size calculation. I agree that the numbers are high, but higher is different from significantly higher

- What was the numbers of ACS in 2019 and 2018?

- Were the episodes of TTS occurred during the time of COVID stay for all the patient ? Implying COVID infection causing TTS vs occurred during the time in month? implying stress and other factors

- What has been the trend of TTS in Italy? Could this trend be suggestive of increase in case detection, earlier diagnosis with each year

- Discuss about the outcomes in the TTS patients, refer and cite PMID: 33759445

Author Response

In this article authors have described the increased in  TTS during the COVID 19 outbreak. I have the following questions.

- Describe what is the baseline ACS admission rates in northen Italy. Consider referring to this article that discussed about reduction in ACS admissions during COVID 19 from Northen Italy, implying that the real numbers could be higher "N Engl J Med 2020; 383:88-89  DOI: 10.1056/NEJMc2009166"

Answer: We thank the reviewer for the observations. We have added the admission rated for ACS in Veneto Region before the pandemic and discussed the issue suggested with the relative reference.

- In the introduction mention the hypothesis, did the authors want to study change in the pattern of TTS during COVID 19.

Answer: Yes, the aim of our hypothesis-generating study was to identify potential influences of the two major pandemic waves of COVID-19 in Italy in the incidence of TTS, also considering the lockdown as potential stressor.

- Why did the authors compare 2020 to 2019+2018 ? If not so clarify

Answer: As written in the text, we analyzed the incidence of TTS over the first and second pandemic waves (2020) and compared it to the same days in 2019 and 2018 years. The choice of 2019+2018 was made in order to obtain a more homogeneous group as reference.

- Include inclusion and exclusion criteria for the study participants

Answer: We have added the inclusion and exclusion criteria as requested.

- How was data extracted? Each hospital vs Private data base, Who extracted the data? Trained health care workers vs insurance/billing companies. Who analyzed the data (Statistician vs trained physician)

Answer: We have added the required information into the methods.

- 32 is 3.9 % for total ACS of 807 not 4.4 %

Answer: We have revised the data. We thank the reviewer for the correction.

- How can the authors say the numbers were significantly high without establishing prior sample size calculation. I agree that the numbers are high, but higher is different from significantly higher

Answer: The incidence of TTS for the previous years was referred to the considered periods for the analysis and not for the entire year.

- What was the numbers of ACS in 2019 and 2018?

Answer: We reviewed the regional heart care report: during the 2018 and 2019 the number of hospitalized ACS were 6.648 and 6.848, respectively.

- Were the episodes of TTS occurred during the time of COVID stay for all the patient ? Implying COVID infection causing TTS vs occurred during the time in month? implying stress and other factors

Answer: As cited into the results., only 3 patients concomitantly had COVID-19 infection and TTS. The other patients experienced TTS during the COVID-19 first and second waves in 2020, without infections.

- What has been the trend of TTS in Italy? Could this trend be suggestive of increase in case detection, earlier diagnosis with each year

Answer: We agreed with the reviewer that the observed trend might be suggestive of increase in case detection, but, to our knowledge, the trend of TTS in Italy is not known. On the other hand, all the European countries (and Italy at the most) were hit by pandemic. Therefore, no “true” reference are available. This point has been addressed in the “limitations section” of the manuscript

- Discuss about the outcomes in the TTS patients, refer and cite PMID: 33759445

Answer: We have added the suggested reference into the discussion and cited it.

Reviewer 2 Report

This is an interesting brief report on a less known and relatively rare syndrome, analyzing its occurrence before and throughout the COVID-19 pandemic. Some suggestion for further clarification below:

-          Spelling of SARS-CoV-2 should be corrected throughout the manuscript – there are multiple spelling mistakes, e.g., on lines 39, 163, 220.

-          Line 40: What is meant by “work lockdown”?

-          Lines 41, 66: While the “major emotional distress” can indeed be a cause of TTS, medical causes, i.e., acute illness (potentially including COVID-19) could also represent triggering factors. Therefore, the abstract and the introduction section of the main manuscript should not only focus on the emotional distress.

-          Line 42: TTS should be written in full.

-          Line 45: How many waves?

-          Lines 45-46: These two phrases were probably meant to be condensed into a single phrase.

-          Line 64: A brief definition of TTS should be provided in the introduction, since this is not actually a very well-known syndrome.

-          Line 67: What is the direct relevance of the economic consequences?

-          Line 95: Please correct “Continuous”.

-          All p values should be provided with 3 decimals, throughout the manuscript and the table.

-          Table: A legend should explain what “Positive COVID-19” refers to. Positive when? Currently? Past xx weeks/months? Ever? Or does this actually reflect whether COVID-19 was considered the triggering factor?

-          Table: The abbreviation CMR should be defined.

-          Table: The table legend should describe the statistical tests applied.

-          Table: It is not enough to report only the p value. Depending on the statistical tests that were performed, other parameters should also be reported, i.e., Chi, df, OR, 95%CI, t, df, U, z, etc. Please check statistical reporting guidelines for each type of test.

-          Line 126: “32 consecutive patients” sounds a bit confusing, since most likely you did not have 32 consecutive patients with TTS, but these were rare occurrences among other patients. The term “consecutive” would be more suited to be moved to the Methods section, where it would thus clarify that you selected all consecutive patients that met the TTS diagnostic criteria.

-          Fig 1B. There was also an increase from 2018 to 2019. Did you check for statistical significance for 2018 vs 2019? Also, what would be the hypothesized explanation for this increase? Could it be better case awareness and the availability of diagnostic criteria rather than an actual increase in the cases? Could this also be the case for 2021? Please discuss this in the manuscript.

-          Please check information regarding troponin on lines 136-137, 168, as it contradicts the data in the Table.

-          Table: How would you explain the discrepancy between troponin T and I in the two groups, with one being higher in the first group and one in the second, both, statistically significant?

-          Line 190: What do you mean by “ all”? The three cases you referenced earlier in the same paragraph?

-          Line 198: The addressability to the hospital for many different types of medical emergencies significantly decreased during 2020 (PMID 35455557).

-          Figure 2: Could you also provide a map of the COVID cases during the first and second wave? This would allow a better visualization to see whether COVID and TTS overlap.

Author Response

This is an interesting brief report on a less known and relatively rare syndrome, analyzing its occurrence before and throughout the COVID-19 pandemic. Some suggestion for further clarification below:

-          Spelling of SARS-CoV-2 should be corrected throughout the manuscript – there are multiple spelling mistakes, e.g., on lines 39, 163, 220.

Answer: We thanks the Reviewer. We have modified the manuscript accordingly.

-          Line 40: What is meant by “work lockdown”?

Answer: We thank the Reviewer for the comment. We have clarified the sentence.

-          Lines 41, 66: While the “major emotional distress” can indeed be a cause of TTS, medical causes, i.e., acute illness (potentially including COVID-19) could also represent triggering factors. Therefore, the abstract and the introduction section of the main manuscript should not only focus on the emotional distress.

Answer: We completely agree with the reviewer. We have modified the paper according to the provided suggestions.

-          Line 42: TTS should be written in full.

Answer: Corrected

-          Line 45: How many waves?

Answer: During the two major waves. We have better specified this aspect as suggested.

-          Lines 45-46: These two phrases were probably meant to be condensed into a single phrase.

Answer: We have modified the sentence according to the suggestion provided by the Reviewer.

-          Line 64: A brief definition of TTS should be provided in the introduction since this is not actually a very well-known syndrome.

Answer: We have provided a short definition of TTS.

-          Line 67: What is the direct relevance of the economic consequences?

Answer: Pandemics cause both short-term fiscal impact and a long-term economic consequence as well as disruption of transportation, shut down of workplaces, restricted trade and travel, and closed land border. All these aspects are reasons for the pandemic's economic slowdown which cause emotional stress in the general population

-          Line 95: Please correct “Continuous”.

Answer: Corrected

-          All p values should be provided with 3 decimals, throughout the manuscript and the table.

Answer: We have provided the third decimal for each p value

-          Table: A legend should explain what “Positive COVID-19” refers to. Positive when? Currently? Past xx weeks/months? Ever? Or does this actually reflect whether COVID-19 was considered the triggering factor?

Answer: We have modified the table for major clarity.

-          Table: The abbreviation CMR should be defined.

Answer: Corrected

-          Table: The table legend should describe the statistical tests applied.

Answer: The statistical test adopted have been described in the statistical section according to the normality (or not) of data distribution.  It's not customary to report into a table, for each raw, the test adopted; moreover, this aspect was not requested by the author’s guidelines.

-          Table: It is not enough to report only the p value. Depending on the statistical tests that were performed, other parameters should also be reported, i.e., Chi, df, OR, 95%CI, t, df, U, z, etc. Please check statistical reporting guidelines for each type of test.

Answer: Ho aggiunto come abbiamo valutato il trend delle ospedalizzazioni nella sezione metodi…altro non manca.

-          Line 126: “32 consecutive patients” sounds a bit confusing, since most likely you did not have 32 consecutive patients with TTS, but these were rare occurrences among other patients. The term “consecutive” would be more suited to be moved to the Methods section, where it would thus clarify that you selected all consecutive patients that met the TTS diagnostic criteria.

Answer: The term consecutive has been used only into the method section as suggested.

-          Fig 1B. There was also an increase from 2018 to 2019. Did you check for statistical significance for 2018 vs 2019? Also, what would be the hypothesized explanation for this increase? Could it be better case awareness and the availability of diagnostic criteria rather than an actual increase in the cases? Could this also be the case for 2021? Please discuss this in the manuscript.

Answer: Not statistical differences were observed comparing 2018 and vs 2019. We have added this result into the legend of Fig. 1B for major clarity. We have added a sentence into the discussion regarding the mentioned non-significant statistical differences.

-          Please check information regarding troponin on lines 136-137, 168, as it contradicts the data in the Table.

Answer: We have revised the data and the Table 1. Now, the rows regarding cardiac troponins (T and I) are correct. We have also revised the text and discussion accordingly. We thank the reviewer for the comment. Troponin levels were higher (both T and I) in the 2020 cohort.

-          Table: How would you explain the discrepancy between troponin T and I in the two groups, with one being higher in the first group and one in the second, both, statistically significant?

Answer: Please see the previous answer. We have revised the data in Table 1. Troponins were higher (both T and I) in the 2020 cohort.

-          Line 190: What do you mean by “all”? The three cases you referenced earlier in the same paragraph?

Answer: Yes. We have revised the sentence for major clarity.

-          Line 198: The addressability to the hospital for many different types of medical emergencies significantly decreased during 2020 (PMID 35455557).

Answer: We agree with the reviewer, but we referred to TTS and therefore to ACSs (not to other disease).

-          Figure 2: Could you also provide a map of the COVID cases during the first and second wave? This would allow a better visualization to see whether COVID and TTS overlap.

Answer: We can’t provide the map required since the available data regarding the distribution of COVID-19 cases are referred to the Health unit and not for cities, towns, etc.

Reviewer 3 Report

Interesting observational research related to a not very well known topic

The article is well written

Author Response

Interesting observational research related to a not very well known topic. The article is well written

Answer: We thank the reviewer for the comments provided.

Reviewer 4 Report

report an increase of takotsubo syndrome (TTS) hospitalization rate  in a large retrospective multicentric register during the 2020 COVID-19 pandemic waves in comparison with a historical 2018-2919 control time period. Although statistically significant, the presumed increase of TTS during the COVID-19 pandemic does not seem clinically relevant, leading to a futile conclusion. In particular, Figure 1 reports higher hospitalization fluctation during the historical control time period, independent of the COVID epidemic.

Author Response

Report an increase of takotsubo syndrome (TTS) hospitalization rate in a large retrospective multicentric register during the 2020 COVID-19 pandemic waves in comparison with a historical 2018-2919 control time period. Although statistically significant, the presumed increase of TTS during the COVID-19 pandemic does not seem clinically relevant, leading to a futile conclusion. In particular, Figure 1 reports higher hospitalization fluctation during the historical control time period, independent of the COVID epidemic.

Answer: We thank the reviewer for the comments. As known, a certain fluctuation regarding the incidence of a disease is common when comparing consecutive years. However, we did not find any statistically significant difference comparing the incidence of TTS between 2018 and 2019. The increasing, albeit not significant trend, observed during the first couple of years of enrollment may be due to different reasons, such as population fluctuation or presence of a seasonality, climatic factors, and seasonality of the disease. For example, TTS has a reverse pattern of seasonal variation compared with acute myocardial infarction, with peaks during summer (Looi J-L., Lee M., Grey C., Webster M., To A., Kerr A.J. Seasonal variation in Takotsubo syndrome compared with myocardial infarction: ANZACS-QI 16. N. Z. Med. J. 2018;131(1471):21–29 and Kanaoka K, Okayama S, Terasaki S, Nakano T, Ishii M, Nakai M, Onoue K, Nishimura K, Yasuda S, Tsujita K, Kawakami R, Miyamoto Y, Tsutsui H, Komuro I, Ogawa H, Saito Y. Role of climatic factors in the incidence of Takotsubo syndrome: A nationwide study from 2012 to 2016. ESC Heart Fail. 2020 Oct;7(5):2629-2636. doi: 10.1002/ehf2.12843. Epub 2020 Jul 27. PMID: 32715646; PMCID: PMC7524088). However, the COVID-19 spread has been slowed down by warm weather as for many other respiratory diseases. Therefore, that period was not considered into the analysis since we analyzed data only during the two major pandemic waves which lead to national lockdown. Moreover, the presence of fluctuations in the incidence of a disease may provide clues to etiology and facilitate the planning of appropriate health services.

Reviewer 5 Report

Authors proposed an article with a strongly interesting topic since it is focused on the Takotsubo syndrome during COVID-19 pandemic in Italy (Veneto region). Authors would try to describe the characteristics and, most of all, the trend of hospitalization of a specific disease, as the TTS, among all the hospitalizations for myocardial infarction during the first year of COVID 19 in Italy (Veneto region).

 In this regard, an appropriate evaluation of the trend of TTS hospitalization despite the global reduction of the hospitalization for myocardial infarction during the first wave of COVID-19 could be very important. A further smart idea of Authors is to consider in their study the triggers variable, by recognizing the potential emotional and physical involvement.

Although Authors explicit part of the limitations of this article, especially concerning the retrospective nature, there are other critical points as described as follow:

The main critical point is related to the primary aim to analyze the trends of patients with TTS admitted during the first and second pandemic waves compared to the same days in 2019 and 2018 years.

To this regard, the choice to evaluate the trend of a rare disease (as the TTS) not considering a continuous timeline period of observation but through two separate window of 45 days have a high risk to generate results strongly conditioned by randomness typical of trend and seasonality of historical data.

Moreover, the Authors stress the results for the discussion section. In the results section are shown the number of cases in the 3 years, more specifically, in 2022 where the total amount of new cases were 32 vs 28 and 27 in 2019 and 18 respectively. Basing on this data the discussion summarize the result in favor of an significantly hypothesis of increasing of TTS during the COVID-19 pandemic period. However, as observable in fig 1A with referring to each single 45 days window, during the first COVID-19 wave the number of TTS decrease of 33% against the cases in the same window of 2018 (the highest number of cases among the two control historical years, 10 in 2020 vs 15 in 2018) and during the second COVID 19 wave the total number increase by 10% in comparison to 2019 (the highest number of cases among the two control historical years, 22 in 2020 vs 20 in 2019). This descriptive analysis seems to not exclude the hypothesis that the small difference observed could be considered as a random difference around the true mean of TTS hospitalization cases. Even if a statistical model for historical data will be not used for the analysis, further weakness points of study results are represented by the evaluation without a specific-year population at risk as denominator and without the usa of a more appropriate probabilistic distribution (for example Poisson).

Minor

Could you better describe how Crude incidence x 10000 inhabitants are calculated?
Geographical maps without an appropriate internal standardization (to compare incidence among adjacent areas) and the use of a correct population at risk 
could be misleading from statistical point of view.

-in row 70 and 209 the study hospital number is 14 instead of 13
- Rows from 146 to 149 appear uncorrected since described the results in terms of Re-hospitalization and mortality but are shown as the difference among the two COVID 19 waves (The rate of hospital readmission and in-hospital mortality were similar between the 146 2020 and the historical cohorts (3.1% vs 9.1%, p=0.28 for the first wave period and 0% vs 147 3.6%, p=0.29 for the second wave phase respectively.).

...

Since 2 years were passed from November 2020, could you perform an update (at least concerning what was happened in 2021) to the dataset?

The first step to analyze the trend require a continuous temporal analysis, could you perform the hospitalization trend data analysis of TTS cases of the entire years and using the 45 days waves or the simply months as covariates in the model?

I strongly recommend the involvement  of a qualified Statistician in the research team.

Author Response

Authors proposed an article with a strongly interesting topic since it is focused on the Takotsubo syndrome during COVID-19 pandemic in Italy (Veneto region). Authors would try to describe the characteristics and, most of all, the trend of hospitalization of a specific disease, as the TTS, among all the hospitalizations for myocardial infarction during the first year of COVID 19 in Italy (Veneto region).

 In this regard, an appropriate evaluation of the trend of TTS hospitalization despite the global reduction of the hospitalization for myocardial infarction during the first wave of COVID-19 could be very important. A further smart idea of Authors is to consider in their study the triggers variable, by recognizing the potential emotional and physical involvement.

Although Authors explicit part of the limitations of this article, especially concerning the retrospective nature, there are other critical points as described as follow:

The main critical point is related to the primary aim to analyze the trends of patients with TTS admitted during the first and second pandemic waves compared to the same days in 2019 and 2018 years.

To this regard, the choice to evaluate the trend of a rare disease (as the TTS) not considering a continuous timeline period of observation but through two separate window of 45 days have a high risk to generate results strongly conditioned by randomness typical of trend and seasonality of historical data.

Answer: We perfectly agree with the reviewer. We have added these aspects as further limitations of the study.

Moreover, the Authors stress the results for the discussion section. In the results section are shown the number of cases in the 3 years, more specifically, in 2022 where the total amount of new cases were 32 vs 28 and 27 in 2019 and 18 respectively. Basing on this data the discussion summarize the result in favor of an significantly hypothesis of increasing of TTS during the COVID-19 pandemic period. However, as observable in fig 1A with referring to each single 45 days window, during the first COVID-19 wave the number of TTS decrease of 33% against the cases in the same window of 2018 (the highest number of cases among the two control historical years, 10 in 2020 vs 15 in 2018) and during the second COVID 19 wave the total number increase by 10% in comparison to 2019 (the highest number of cases among the two control historical years, 22 in 2020 vs 20 in 2019). This descriptive analysis seems to not exclude the hypothesis that the small difference observed could be considered as a random difference around the true mean of TTS hospitalization cases. Even if a statistical model for historical data will be not used for the analysis, further weakness points of study results are represented by the evaluation without a specific-year population at risk as denominator and without the usa of a more appropriate probabilistic distribution (for example Poisson).

Answer: We thank the reviewer for the observations. We performed a hypothesis-generating study using the available data collected during the first two pandemic waves of COVID-19 infection in Italy. As the reviewer can image, it is difficult to collect data in such circumstances. Therefore, our hypothesis must be confirmed by further, larger and multicentric studies. We have added also this second statistical drawback to the limitations of the study, also highlighting the need for further studies to confirm our results.

Minor

Could you better describe how Crude incidence x 10000 inhabitants are calculated?

Answer: Crude Rate was defined as (Number of Events t ÷ Mid-year Population) x 10000

Geographical maps without an appropriate internal standardization (to compare incidence among adjacent areas) and the use of a correct population at risk could be misleading from statistical point of view.

Answer: We thank the reviewer for the insightful comment. Unfortunately, our study was retrospective, observational and hypothesis generating based on the available data. Moreover, Veneto region and the northeastern Italy are quite homogeneous in term of demographical, economical, and social characteristics

-in row 70 and 209 the study hospital number is 14 instead of 13

Answer: The entire manuscript has been revised. We thank the reviewer for the comment

- Rows from 146 to 149 appear uncorrected since described the results in terms of Re-hospitalization and mortality but are shown as the difference among the two COVID 19 waves (The rate of hospital readmission and in-hospital mortality were similar between the 146 2020 and the historical cohorts (3.1% vs 9.1%, p=0.28 for the first wave period and 0% vs 147 3.6%, p=0.29 for the second wave phase respectively.).

Answer: in this section we report the outcome (re-hospitalization + in-hospital mortality) comparing the incidence between 2020 and historical cohorts both during the first and the second wave of pandemic. We found no significant difference both during the first and the second wave of pandemic when the 2020 cohort was compared to historical cohorts.

Since 2 years were passed from November 2020, could you perform an update (at least concerning what was happened in 2021) to the dataset?

Answer: Unfortunately, 2021 data are not available. However, it is noteworthy that the pandemic hit Italy, both in term of COVID 19 mortality and morbidity and of economic and social burden, stronger in 2020 than in 2021.

The first step to analyze the trend require a continuous temporal analysis, could you perform the hospitalization trend data analysis of TTS cases of the entire years and using the 45 days waves or the simply months as covariates in the model?

Answer: We thank the reviewer for the insightful observation, but, unfortunately, we were not able to run this analysis because of unavailability of the data. We addressed this point in the limitation section. However, it should be noted that COVID-19 pandemic affected the national health system along the entire 2020: therefore, we focused in specific time windows when the spreading of SARS-CoV-2 (first and second waves) was the highest and consequently the social and economic restrictions (i.e. quarantine and lockdown) were the strongest. 

I strongly recommend the involvement of a qualified Statistician in the research team.

Answer: We thank the reviewer for the suggestion

Round 2

Reviewer 4 Report

The Authors of the manuscript report an increase of takotsubo syndrome (TTS) hospitalization rate  in a large retrospective multicentric register during the 2020 COVID-19 pandemic waves in comparison with historical 2018-2919 control time period. Even though statistically significant the presumed increase of TTS during the COVID-19 pandemic seems clinically non relevant leading to futile conclusion. In particolar, the Fig 1 reports a higher hospitalization fluctation during the historical control time period independent of COVID epidemic. 

Author Response

Answer: We understood the comment of the reviewer, but we disagreed about the futility of our conclusions, despite the fluctuation during the historical control time period (which, as said in round 1, is reasonable when comparing consecutive years, especially if some affecting factors, such as seasonality, climatic factors, etc. have been already associated by previous investigations with the disease, as TTS), for several reasons. First, our pre-pandemic data are in line with the crude TTS incidence reported by the analyses of Jabri et al. “Incidence of Stress Cardiomyopathy During the Coronavirus Disease 2019 Pandemic. JAMA, doi: 10.1001/jamanetworkopen.2020.14780” (these results has been added and briefly discussed in this revised version of the paper). Second, regarding the increased number of TTS cases and relative rates, our data are certainly underestimated. Indeed, during the first two pandemic waves, patients with ACS might not seek medical attention due to the fear of becoming infect, resulting in an incorrigible sampling bias. This epidemiological aspect has been largely reported in recent literature and also observed in our setting (Viruses 2022, 14, 1925). Finally, it should be remembered that as retrospective, real world observation, this report is hypothesis-generating study, which requires further confirmations by larger analyses, although other preliminary findings seem to confirm our data. (Shah RM, Shah M, Shah S, Li A, Jauhar S. Takotsubo Syndrome and COVID-19: Associations and Implications. Curr Probl Cardiol. 2021;46:100763 and the previously cited article by Jabri et al).

Reviewer 5 Report

Authors produce many revision to the article improving is quality.

Main critical points were not completely solved but are now specified as part of limitations of the study, giving to the reader a better awareness of results, conclusion and of the manuscript aim intended to help the generating hypothesis process in future and more robust studies.

Only few minors revisions are required now:

1) Concerning lines 125 to 127 in results section : “Compared to the historical cohorts, a significant increase in the number of TTS cases 125 occurred in 2020 (from to 27 to 32, +15.6%, p=0.031 and from 28 to 32, +12.5%, p=0.042, 126 comparing 2018 to 2020 and 2019 to 2020, respectively)” since you intend to express the increasing from a historical years, the correct values must be calculated as 5 on 27 and 4 on 28. In fact, + 5 new cases more on 2018 represents an increasing of 18.5% (no +15.6%)  and, respectively in 2019, + 4 are the 14.3% on 2019 (no +12.5%).

2) Concerning before comments:
“- Rows from 146 to 149 appear uncorrected since described the results in terms of Re-hospitalization and mortality but are shown as the difference among the two COVID 19 waves (The rate of hospital readmission and in-hospital mortality were similar between the 2020 and the historical cohorts (3.1% vs 9.1%, p=0.28 for the first wave period and 0% vs 3.6%, p=0.29 for the second wave phase respectively.).”

Answer: in this section we report the outcome (re-hospitalization + in-hospital mortality) comparing the incidence between 2020 and historical cohorts both during the first and the second wave of pandemic. We found no significant difference both during the first and the second wave of pandemic when the 2020 cohort was compared to historical cohorts.

In lines 165 to 166 :”(3.1% vs 9.1%, p=0.281 for the first wave period and 0% vs 3.6%, p=0.292 for the second wave phase, respectively. The total in-hospital stay was shorter in 2020 compared to the historical cohorts (p=0.021).” the sentece is misleading  since “3.1% vs 9.1%, p=0.281 is not the results “for the first wave period” but, according to table 1,  it is the results in term of comparison of Re-hospitalization global 2020 ( sum of cases observed in the two 2020 waves) vs  the sum of cases observed in the two past years(historical cohorts). In the same way, 0% vs 3.6%, p=0.292  is not the results “for the second wave phase, respectively”  but it is the results of comparison of  global 2020 ( sum of cases in the two 2020 waves) vs  the two 2 years( historical cohorts )in terms of  in-hospital mortality.

Moreover, in these lines a closed parenthesis “)” is missing and the p.value reported in the text are different from table 1 (0.281 vs 0.282 and 0.296 vs 0.292).

3) In the limitation paragraph 4.1, to sustain the  second limitation cite  Caminiti et al. (  Effects of the COVID-19 Epidemic on Hospital Admissions for Non-Communicable Diseases in a Large Italian University-Hospital: A Descriptive Case-Series Study )in which in a similar area  (Parma, Italy)  was analyzed the change in terms of number of hospitalization and the potential role played by the reallocation of healthcare resources. I also suggest to cite Shah et al. (Takotsubo Syndrome and COVID-19: Associations and Implications) that analyze the increasing of TTS and the possible association between stress-induced cardiomyopathy and COVID-19 (in both the general population and in COVID-19 patients)

4) In lines 112 to 116 was added into the text that a Poisson regression model was now performed. However, I do not find in result section (or in supplementary material) the aforesaid model .

Author Response

Authors produce many revision to the article improving is quality.

Main critical points were not completely solved but are now specified as part of limitations of the study, giving to the reader a better awareness of results, conclusion and of the manuscript aim intended to help the generating hypothesis process in future and more robust studies.

Only few minors revisions are required now:

1) Concerning lines 125 to 127 in results section : “Compared to the historical cohorts, a significant increase in the number of TTS cases 125 occurred in 2020 (from to 27 to 32, +15.6%, p=0.031 and from 28 to 32, +12.5%, p=0.042, 126 comparing 2018 to 2020 and 2019 to 2020, respectively)” since you intend to express the increasing from a historical years, the correct values must be calculated as 5 on 27 and 4 on 28. In fact, + 5 new cases more on 2018 represents an increasing of 18.5% (no +15.6%)  and, respectively in 2019, + 4 are the 14.3% on 2019 (no +12.5%).

Answer: We thank the reviewer for the comments. We have revised the results and provided adequate corrections.

2) Concerning before comments:
“- Rows from 146 to 149 appear uncorrected since described the results in terms of Re-hospitalization and mortality but are shown as the difference among the two COVID 19 waves (The rate of hospital readmission and in-hospital mortality were similar between the 2020 and the historical cohorts (3.1% vs 9.1%, p=0.28 for the first wave period and 0% vs 3.6%, p=0.29 for the second wave phase respectively.).”

“Answer: in this section we report the outcome (re-hospitalization + in-hospital mortality) comparing the incidence between 2020 and historical cohorts both during the first and the second wave of pandemic. We found no significant difference both during the first and the second wave of pandemic when the 2020 cohort was compared to historical cohorts.”

In lines 165 to 166 :”(3.1% vs 9.1%, p=0.281 for the first wave period and 0% vs 3.6%, p=0.292 for the second wave phase, respectively. The total in-hospital stay was shorter in 2020 compared to the historical cohorts (p=0.021).” the sentece is misleading  since “3.1% vs 9.1%, p=0.281 is not the results “for the first wave period” but, according to table 1,  it is the results in term of comparison of Re-hospitalization global 2020 ( sum of cases observed in the two 2020 waves) vs  the sum of cases observed in the two past years(historical cohorts). In the same way, 0% vs 3.6%, p=0.292  is not the results “for the second wave phase, respectively”  but it is the results of comparison of  global 2020 ( sum of cases in the two 2020 waves) vs  the two 2 years( historical cohorts )in terms of  in-hospital mortality.

Answer: We thank again the reviewer for the valuable suggestions. We have rephrased the paragraph for major clarity.

The global rate of hospital readmission observed in 2020, defined as the sum the cases observed during the two pandemic waves in that year) as well as the overall mortality rate in the same period were similar when compared to the historical cohorts

Moreover, in these lines a closed parenthesis “)” is missing and the p.value reported in the text are different from table 1 (0.281 vs 0.282 and 0.296 vs 0.292).

Answer: We thank the reviewer again. We have corrected the typing errors.

3) In the limitation paragraph 4.1, to sustain the  second limitation cite  Caminiti et al. (  Effects of the COVID-19 Epidemic on Hospital Admissions for Non-Communicable Diseases in a Large Italian University-Hospital: A Descriptive Case-Series Study )in which in a similar area  (Parma, Italy)  was analyzed the change in terms of number of hospitalization and the potential role played by the reallocation of healthcare resources. I also suggest to cite Shah et al. (Takotsubo Syndrome and COVID-19: Associations and Implications) that analyze the increasing of TTS and the possible association between stress-induced cardiomyopathy and COVID-19 (in both the general population and in COVID-19 patients)

Answer: We have added the suggested references into the manuscript and briefly discussed the paper published by Shah et al.  

4) In lines 112 to 116 was added into the text that a Poisson regression model was now performed. However, I do not find in result section (or in supplementary material) the aforesaid model.

Answer: We have better defined the statistical methods. Poisson regression was used to estimate the annual percentage change (AAPC) in TTS events with 95% confidence interval (CI). These results have been presented into a new table (New table 1). We believe that the significant AAPC further support our preliminary findings.